

# Transcriptomic study of the mechanism of anoikis resistance in head and neck squamous carcinoma

Chen Guo[1,*], Ling-feng Xu[1,*], Hui-min Li[1], Wei Wang[1], Ji-hua Guo[1], Meng-qi Jia[2], Rong Jia[1] and Jun Jia[1,2]

[1] The State Key Laboratory Breeding Base of Basic Science of Stomatology (Hubei-MOST) & Key Laboratory of Oral Biomedicine Ministry of Education, Wuhan University, School and Hospital of Stomatology, Wuhan, Hubei, China

[2] Department of Oral and Maxillofacial Surgery, Wuhan University, School and Hospital of Stomatology, Wuhan, Hubei, China

[*] These authors contributed equally to this work.

## ABSTRACT

**Background**. Normal epithelial cells rapidly undergo apoptosis as soon as they lose contact with the extracellular matrix (ECM), which is termed as anoikis. However, cancer cells tend to develop a resistance mechanism to anoikis. This acquired ability is termed as anoikis resistance. Cancer cells, with anoikis resistance, can spread to distant tissues or organs via the peripheral circulatory system and cause cancer metastasis. Thus, inhibition of anoikis resistance blocks the metastatic ability of cancer cells.

**Methods**. Anoikis-resistant CAL27 (CAL27$^{AR}$) cells were induced from CAL27 cells using the suspension culture approach. Transcriptome analysis was performed using RNA-Seq to study the differentially expressed genes (DEGs) between the CAL27$^{AR}$cells and the parental CAL27 cells. Gene function annotation and Gene Ontology (GO) enrichment analysis were performed using DAVID database. Signaling pathways involved in DEGs were analyzed using Gene Set Enrichment Analysis (GSEA) software. Analysis results were confirmed by reverse transcription PCR (RT-PCR), western blotting, and gene correlation analysis based on the TCGA database.

**Results**. GO enrichment analysis indicated that the biological process (BP) of the DEGs was associated with epidermal development, DNA replication, and G1/S transition of the mitotic cell cycle. The analysis of cellular component (CC) showed that the most significant up-regulated genes were related to extracellular exosome. KEGG Pathway analysis revealed that 23 signaling pathways were activated ($p$-value $\leq$ 0.05, FDR $q$-value $\leq$ 0.05) and 22 signaling pathways were suppressed ($p$-value $\leq$ 0.05, FDR $q$-value $\leq$ 0.05). The results from the GSEA indicated that in contrast to the inhibition of EGFR signaling pathway, the VEGF signaling pathway was activated. The VEGF signaling pathway possibly activates STAT3 though induction of STAT3 phosphorylation. Gene correlation analysis revealed that the VEGFA- STAT3-KLF4-CDKN1A signal axis was not only present in head and neck squamous carcinoma (HNSCC) but also two other epithelial-derived carcinomas that highly express VEGFA, including kidney renal clear cell carcinoma (KIRC) and ovarian serous cystadenocarcinoma (OV).

Corresponding authors
Rong Jia, jiarong@whu.edu.cn
Jun Jia, junjia@whu.edu.cn

# INTRODUCTION

In 2012, 529,500 patients suffered from lip, oral cavity, and pharyngeal cancers globally, accounting for 3.8% of all cancer cases. It is predicted that by 2035, the incidence of lip, oral and pharyngeal cancer will increase by 62%, reaching to 856,000 cases (*Shield et al., 2017*). Head and neck squamous cell carcinoma (HNSCC), which is mainly consisted of cancers from lip, oral cavity, and pharynx, accounts for 90% of head and neck cancers (*Suh et al., 2014*). Noticeably, there were several reports that claimed that the long-term prognosis of many patients with HNSCC is poor due to cancer distant metastasis (*Li et al., 2017*; *Wreesmann et al., 2004*). Cancer distant metastasis includes four main steps: invasion, intravasation, dissemination and colonization (*Su et al., 2015*). The prerequisite of dissemination is that cancer cells could acquire the ability to resist anoikis and survive in the lymphatic and circulatory system (*Yan et al., 2005*). Anoikis is a kind of cell programmed death which occurs as a result of cell detachment from the extracellular matrix (ECM). For this reason, metastasis is inefficient when cancer cells undergo anoikis (*Kim et al., 2012b*). Unfortunately, a few cancer cells are likely to develop the ability to resist anoikis, which helps them to survive in the circulatory system (*Simpson, Anyiwe & Schimmer, 2008*).

There are several mechanisms that endow cancer cells with the characteristics of anoikis resistance, including the following: (1) the activation of the anti-apoptotic signaling pathways, such as integrin signaling pathway and PI3K/Akt signaling pathway, which allows transformed cancer cells to survive in the circulatory system (*Attwell, Roskelley & Dedhar, 2000*; *Davies et al., 1999*; *Vitolo et al., 2009*); (2) secretion of autocrine growth factors, such as EDIL3 and VEGFA, which aids proliferation, survival and migration (*Feng et al., 2014*; *Sher et al., 2009*); and (3) overexpression of Twist, HGF/Met and NF-kB through epithelial mesenchymal transformation (EMT) (*Thompson, Newgreen & Tarin, 2005*). Research relating to the mechanisms contributing to anoikis resistance can provide us with an opportunity to limit anoikis resistance, which would prevent cancer cell metastasis.

Many studies aimed to reveal the mechanism of anoikis resistance in HNSCC (*Liao et al., 2017*; *Shen et al., 2017*; *Yadav et al., 2011*; *Zeng et al., 2002*). EMT phenomenon is widely accepted as one the most important factors for cancer cells acquiring anoikis resistance (*Taddei et al., 2012*). Transcriptional profiling has identified that the strongly upregulated transcripts include genes that are reportedly involved in invasion and metastasis, such as Tbx3, DOCK10, LOX, ROBO1 and SRGN, following induction of EMT in HNSCC (*Humtsoe et al., 2012*). Moreover, *Yan et al. (2005)* demonstrated that NF-κB, as a transcriptional regulator of many genes, is associated with inflammatory mediator production and apoptosis. They put forward that detachment could directly activates NF-κB and that the NF-κB activation has positive effect on the early course of anoikis because of inflammation factor stimulation generated by NF-κB activation. Which genes play significant roles in anoikis resistance in HNSCC remains unknown. In this study, we conducted transcriptional studies on anoikis resistance in HNSCC. Anoikis-resistant CAL27[AR] cells were acquired using a suspension culture approach (*Kupferman et al., 2007*), and then RNA-Seq was used to obtain deep sequencing data for whole transcriptome analysis between CAL27 cells and CAL27[AR] cells, which helped us to

obtain a large number of differentially expressed genes (DEGs) between these two kinds of cells. Through bioinformatics analysis and verification of DEGs, we found several signaling pathways related to the ability of anoikis resistance in CAL27$^{AR}$ cells. Furthermore, we analyzed these signaling pathways and found a signal axis related to the ability of anoikis resistance in multiple epithelial tumors that overexpressed VEGFA. Our study, for the first time, reveals the changes in transcriptional level of anoikis-resistant head and neck squamous cell carcinoma cells.

## MATERIAL AND METHODS

### CAL27 cells monolayer culture

Oral squamous cell carcinoma cell line, CAL27, was maintained in adherent culture conditions and was cultivated in Dulbecco's Modified Eagle Medium (DMEM; Hyclone, South Logan, UT, USA) supplemented with 10% fetal bovine serum (FBS, Hyclone) and 1% antibiotic-antimycotic solution in an incubator with moist air containing 5% $CO_2$ at 37 °C.

### Collection for CAL27 cells resisting to anoikis

To collect anoikis-resistant CAL27 cells, according to previously reported procedures by *Kupferman et al. (2007)*, we digested approximately $2 \times 10^8$ CAL27 cells using 0.25% trypsin with 0.1% EDTA solution from a monolayer grown in a tissue culture flask. Then, the CAL27 cells were transferred to a 10% poly-HEMA (Sigma Chemical Co. USA) coated tissue culture dish (10 cm) for 72 h in an incubator with moist air containing 5% $CO_2$ at 37 °C. Subsequently, these CAL27 spheroids cells were centrifuged at 300 rpm for 5 min. Then, they were transferred to an uncoated dish, and were allowed to replicate for 24 h. After six rounds of coated and un-coated culture, anoikis-resistant CAL27 cells were harvested at 300 rpm for 5 min; they were marked as CAL27$^{AR}$ cells.

### RNA-seq and data analysis

Total RNA was isolated from the three CAL27 samples and three CAL27$^{AR}$ samples using TRIzol (Invitrogen, Carlsbad, CA, USA) according to the manufacturer's instructions. The concentration and quality of RNA samples were determined using a Nano Drop 2000 micro-volume spectrophotometer (Thermo Scientific, Waltham, MA, USA). Library construction and sequencing were performed by the BGI Genomics Institute, Wuhan, China (https://www.bgi.com/). The library preparation was followed by BGI's standard procedure. The libraries were sequenced on the Illumina HiSeq 2000 platform using the 50-bp pair-end sequencing strategy and 60 million reads were generated for each sample. DEGs screening aimed to find DEGs among samples and perform further function analysis on them. We used the Noiseq package method for screening analysis of DEGs. First, Noiseq used sample's gene expression in each group to calculate log2 (foldchange) M and absolute different value D of all pair conditions to build noise distribution model. Second, for gene A, Noiseq computes its average expression "Control-avg" in control group and average expression "Treat-avg" in treatment group. Then the foldchange (MA = log2((Control-avg)/(Treat-avg))) and absolute different value D (DA = |Congrol-avg-Treat-avg|) will be

obtained. If MA and DA markedly diverge from the noise distribution model, gene A will be defined as a DEG. "Loget" represents log2 fold change (*Pacholewska, 2017*). "Probability" represents probability of differential expression. Noiseq obtains probability to assess the reliability of the difference (*Tarazona et al., 2011*). In order to visualize the distribution of DEGs among samples, cluster analysis of differentially expressed genes was carried out. We chose the top DEGs ($|Loget| \geq 3$, probability $\geq 0.9$) for display. Heatmapping of top DEGs was generated by OmicShare tools (http://www.omicshare.com/tools/). The raw data have been deposited at the National Center for Biotechnology Information Sequence Read Archive (SRA) with associated accession numbers SRP158985. The workflow of data analysis was summarized in Fig. S1.

## Database for Annotation, Visualization and Integrated Discovery (DAVID) functional annotation analysis

To perform gene ontology (GO) enrichment analysis, DEGs ($|Loget| \geq 2$, probability $\geq 0.9$) were explored with DAVID functional annotation tool (http://david.abcc.ncifcrf.gov) (*Huang, Sherman & Lempicki, 2008*; *Huang, Sherman & Lempicki, 2009*). Among the enrichment results of up- or down-regulated DEGs, the top ten statistically significant enriched terms ($p$-value $\leq 0.05$ and FDR $\leq 0.05$) of biological processes, molecular function and cellular component, as well as three ontologies of GO, were picked up separately. Regarding the visualization of GO enrichment analysis results, the top ten statistically significant enriched terms were visualized using an online tool (http://www.ehbio.com). We further used DAVID to re-analyze the genes enriched in the terms of "protein binding" and up- or down-regulated genes enriched in the "VEGFA_ UP.V1_UP" and "VEGFA_ UP.V1_DN" terms. $p$-value $\leq 0.05$ and FDR $\leq 0.05$ was considered as statistically significant.

## Gene Set Enrichment Analysis (GSEA)

Activated or repressive pathways were detected using Gene Set Enrichment Analysis (GSEA) (GSEA Desktop v3.0, Broad Institute) (http://www.broadinstitute.org/gsea/index.jsp) (*Mootha et al., 2003*; *Subramanian et al., 2005*). Briefly, the "number of permutations" was set to 1,000, the "collapse data set to gene symbols" was set to False, the "enrichment statistic" was set to Weighted, the "metric for ranking genes" was set to Singal2Noise, and the "permutation type" was set to gene set. The Molecular Signatures Database (MSigDB) collected a large quantity of annotated gene sets for analysis with the GSEA software. MSigDB gene sets are divided into eight major collections. C2 collection and C6 collection in the MSigDB were used to detect activated or repressive pathways related to our DEGs, respectively. The used gene sets for enrichment analysis included "CP: KEGG: KEGG gene sets" (gene sets derived from the KEGG pathway database.), "CP: REACTOME: Reactome gene sets" (gene sets derived from the Reactome pathway database) and "C6: oncogenic signatures" (gene sets that represent signatures of cellular pathways). The $p$-value $\leq 0.05$ and FDR $q$-value $\leq 0.05$ was considered as statistically significant.

## Gene expression analysis

VEGFA gene expression analysis in different types of tumors was done using GEPIA, an online tool for analyzing the RNA sequencing expression data from TCGA (http://gepia.cancer-pku.cn/index.html) (*Tang et al., 2017*). The steps were as follows: The "Gene Expression Profile" tool was used; "VEGFA" was input into the "Gene column"; "Differential Methods" was set to "ANOVA"; " |Log2FC| Cutoff" was set to "1"; " *p*-value Cutoff" was set to "0.01"; "Log Scale" was set to "No"; "Matched Normal data" was set to "Match TCGA normal and GTEx data". All types of cancer names, except mesothelioma and uveal melanoma, from the "Dataset" was added to "Tissue Order". To better demonstrate the expression of VEGFA in tumor tissues and normal tissues, we also created a box plot for the expression of VEGFA. Briefly, the "BoxPlot" tool was selected, and "VEGFA" was inputted into the "Gene", " |Log2FC| Cutoff" was set to "1", " *p*-value Cutoff" was set to "0.01", "Log Scale" was set to "YES", and HNSC, KIRC, OV and GBM were added into "Datasets".

## Gene correlation analysis

Gene correlation analysis was done using GEPIA. The "correlation coefficient" was set to "Spearman". "HNSC Tumor", "KIRC Tumor", "OV Tumor" and "GBM Tumor" were added into the "Used Expression Datasets" column, respectively *p*-value ≤ 0.05 was considered as statistically significant.

## Reverse transcription PCR (RT-RCR)

Total cellular RNA was extracted using a Multisource Total RNA Miniprep Kit (Axygen, Tewksbury, MA, USA). RNA was treated with DNase I (Invitrogen) and reverse-transcribed using random primers (hexadeoxynucleotides) (Promega, Madison, WI, USA) and Moloney Murine Leukemia Virus (MMLV) Reverse Transcriptase (Promega). PCR was performed using rTaq DNA polymerase (TaKaRa, Tokyo, Japan). The primer sequences are listed in the Table S1.

## Western blotting

Protein samples in the 2× SDS sample buffer were denatured by boiling for 5 min in 95 °C, separated by 10% SDS-PAGE gels and transferred to a nitrocellulose membrane. The membrane was blotted with 5% nonfat dry milk for 1 h at room temperature and then probed with dilutions recommended by the primary antibody suppliers overnight at 4 °C, including mouse monoclonal antibodies against STAT3 Tyr 705 (sc-8059 Santa Cruz Biotechology, Santa Cruz, CA, USA), mouse monoclonal antibodies against STAT3 Ser 727 (sc-136193 Santa Cruz, USA) and mouse monoclonal antibodies against GAPDH (sc-47724 Santa Cruz, USA).

# RESULTS

## Cluster analysis of DEGs and RT-PCR confirmation

To identify DEGs between CAL27[AR] cells and CAL27 cells, we illustrated the gene expression features of CAL27[AR] cells and CAL27 cells using RNA sequencing (RNA-seq) technology.

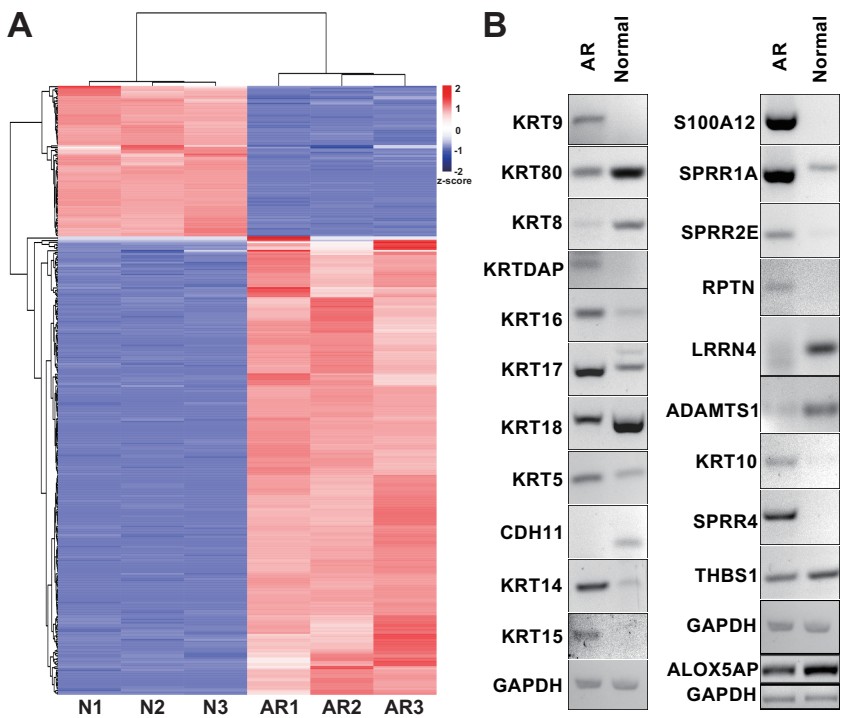

**Figure 1 Cluster analysis of DEGs and RT-PCR confirmation.** The cluster heatmap of DEGs (|Loget| ≥ 3, probability ≥ 0.9) was generated by OmicShare tools (http://www.omicshare.com/tools/). (A) Cluster heatmap of the most significantly changed 360 DEGs (|Loget| ≥ 3, probability ≥ 0.9).The color of the squares in the heatmap reflects the $z$-score. The result of cluster analysis indicated that the samples had low heterogeneity and high repeatability. (B) To verify our RNA-seq results, we selected 21 genes, including 15 keratin-related genes and six other randomly selected genes, for examination by using RT-PCR. GAPDH was selected as control. ALOX5AP has an independent control. All the detected 21 genes matched the results of RNA-Seq. It was revealed that high-throughput sequencing results are reliable.

The number of significantly changed DEGs between CAL27$^{AR}$ and CAL27cells was 2,847 (|Loget| ≥ 1, probability ≥ 0.8), of which 1,558 genes were up-regulated and 1,289 genes were down-regulated in CAL27$^{AR}$ cells (File S1). Furthermore, we performed cluster analysis of the top 360 genes (|Loget| ≥ 3, probability ≥ 0.9) expression patterns. The significantly up-regulated and down-regulated genes are shown in the cluster heatmap (Fig. 1A and Fig. S2). The result of cluster analysis indicated that the samples had low heterogeneity and high repeatability. In addition, we verified the results of RNA-Seq using RT-PCR. All the detected 21 genes, including 15 keratin-related genes and 6 randomly selected other genes, matched the results of RNA-Seq, which confirmed that the high-throughput sequencing results were reliable (Fig. 1B).

## Functional annotation analysis of DEGs was done using Database for Annotation, Visualization and Integrated Discovery (DAVID)

To facilitate the comprehensive analysis of DEGs, total of 487 up-regulated genes and 335 down-regulated genes (|Loget| ≥ 2, probability ≥ 0.9) were integrated to gene ontology (GO) terms using DAVID. The GO classification was divided into three categories:

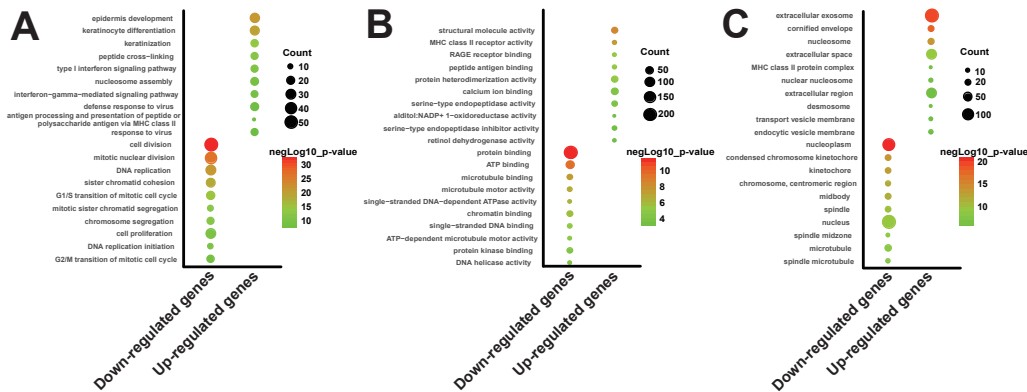

**Figure 2** **Gene ontology enrichment analysis of DEGs.** To facilitate the comprehensive analysis of DEGs, a total of 487 up-regulated genes and 335 down-regulated genes (|Loget| ≥ 2, probability ≥ 0.9) were analyzed using DAVID. (A) The top ten terms of biological process (BP), enriched in up- or down-regulated DEGs. (B) The top ten terms of molecular function (MF), enriched in up- or down-regulated DEGs. (C) The top ten terms of cellular component (CC), enriched in up- or down-regulated DEGs. $p$-value ≤ 0.05, FDR ≤ 0.05.

biological process (BP), molecular function (MF) and cellular component (CC). In each category, top ten significant terms enriched in the up- or down-regulated genes were shown in the Fig. 2. Regarding biological process, many up-regulated genes were associated with "epidermis development" ($p$-value $= 1.01 \times 10^{-21}$), "keratinocyte differentiation" ($p$-value $= 1.62 \times 10^{-20}$) and "keratinization" ($p$-value $= 1.65 \times 10^{-14}$). In contrast, the top three terms in down-regulated genes were "cell division" ($p$-value $= 4.49 \times 10^{-33}$), "mitotic nuclear division" ($p$-value $= 6.38 \times 10^{-27}$) and "DNA replication" ($p$-value $= 1.24 \times 10^{-21}$) (Fig. 2A). The molecular functions of up- or down-regulated genes were closely related to "structural molecule activity" ($p$-value $= 2.31 \times 10^{-9}$) and "protein binding" ($p$-value $= 2.63 \times 10^{-12}$) respectively. Moreover, we noticed that a great number of up-regulated genes were enriched into "protein heterodimerization activity" ($p$-value $= 4.62 \times 10^{-6}$) and "calcium ion binding" ($p$-value $= 4.12 \times 10^{-5}$) (Fig. 2B). Most significantly down-regulated genes were associated with protein binding, however, their biological functions were unclear. For this reason, we further re-analyzed these genes related to protein binding using DAVID. The results revealed that most down-regulated genes related to protein binding could influence "cell cycle" ($p$-value $= 6.27 \times 10^{-21}$), "DNA replication" ($p$-value $= 3.33 \times 10^{-7}$) and "p53 signaling pathway" ($p$-value $= 2.31 \times 10^{-4}$). Interestingly, the results of cellular component indicated that 146 up-regulated genes, including S100A8, S100A9, WNT4 and CD14, were enriched in the "extracellular exosome" term ($p$-value $= 1.02 \times 10^{-20}$). These up-regulated genes related to the "extracellular exosome" term were summarized in Table S2. The results also indicated that 117 down-regulated genes were enriched in the "nucleoplasm" term ($p$-value $= 7.76 \times 10^{-22}$). Besides these positions, many up-regulated gene products were enriched in the extracellular space ($p$-value $= 1.21 \times 10^{-10}$) and extracellular region ($p$-value $= 9.22 \times 10^{-7}$). Furthermore, there were

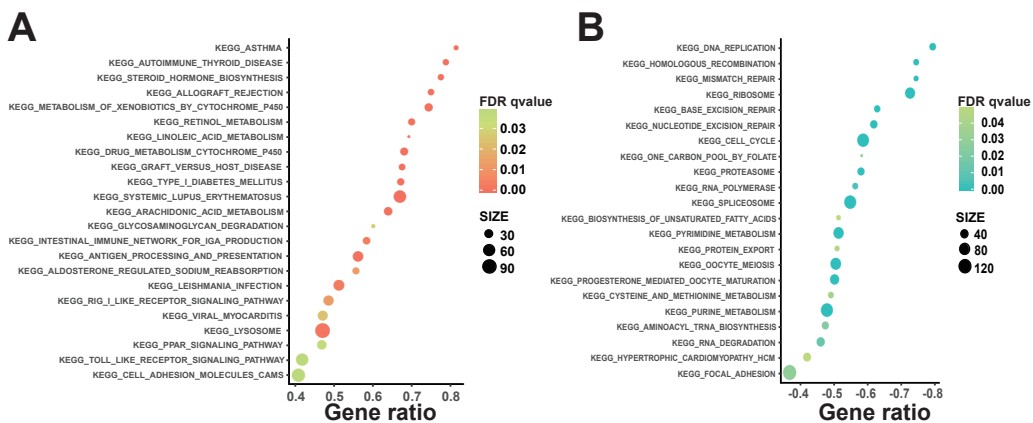

**Figure 3  KEGG pathway analysis of DEGs using GSEA.** To further deliberate signal pathways of DEGs, the KEGG pathway that correlates with the DEGs was studied using GSEA. (A) Activated pathway in CAL27$^{AR}$ cells, including drug metabolism cytochrome P450, PPAR signaling pathway, cell adhesion molecules (Cams), arachidonic metabolism, linoleic metabolism and Toll-like receptor signaling pathway etc. (B) Suppressive pathway in CAL27$^{AR}$ cells, including DNA replication, mismatch repair, nucleotide excision repair and cell cycle pathways etc.

a great number of down-regulated gene products located in the nucleus ($p$-value = $1.10 \times 10^{-10}$) (Fig. 2C). For all results, FDR ≤ 0.05.

## Gene Set Enrichment Analysis (GSEA) of DEGs revealed some important signaling pathways and characteristics of oncogene in CAL27$^{AR}$ cells

To further explore signal pathways of DEGs, the biological significance related with the DEGs was studied using GSEA. Different collections of molecular signatures database (MSigDB) gene sets were used to detect biological significance of DEGs. These collections of MSigDB gene sets included chemical and genetic perturbations (CGP), KEGG gene sets, Reactome gene sets and oncogenic signatures. Activated signaling pathways included drug metabolism cytochrome P450, PPAR signaling pathway, cell adhesion molecules (Cams), arachidonic metabolism, linoleic metabolism and Toll like receptor signaling pathway (Fig. 3A). In addition, several signaling pathways were suppressed in CAL27$^{AR}$ cells, such as DNA replication, mismatch repair, nucleotide excision repair and cell cycle pathways (Fig. 3B). The GSEA of Reactome pathways also indicated that some cell cycle processes and events were suppressed. These cell cycle processes and events included G1/S specific transcription, G0 and early G1, mitotic M/G1 phases, and G1/S transition (Table S3). Furthermore, we found that "VEGF_A_UP.V1_UP" and "P53_DN.V2_UP" gene sets could be enriched in up-regulated genes, and "VEGF_A_UP.V1_DN" gene sets could be enriched in down-regulated genes when the gene sets of oncogenic signatures were used to analyze DEGs (Figs. 4A–4C). The GSEA of CGP disclosed that the "KOBAYASHI_EGFR_SIGNALING_24HR_UP" and "YAN_ESCAPE_FROM_ANOIKIS" gene sets could be enriched in up-regulated genes, and the "KOBAYASHI_EGFR_SIGNALING_24HR_DN" gene set could be enriched in down-regulated genes (Figs. 4D–4F).

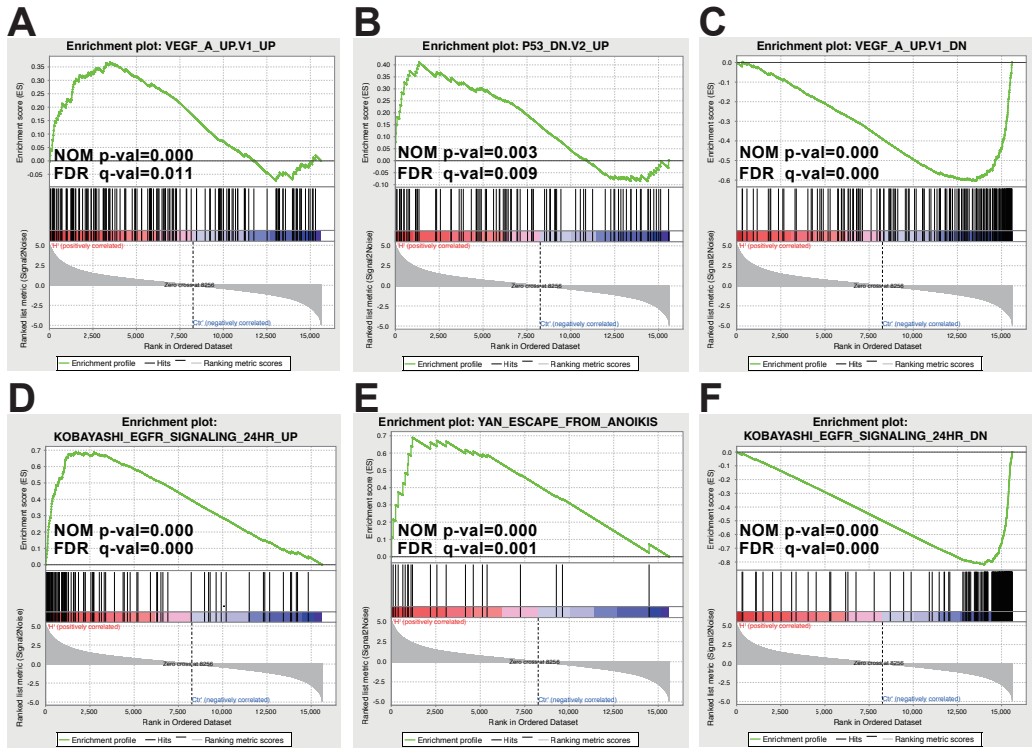

**Figure 4** **Oncogenic signatures and CGP analysis of DEGs using GSEA.** To further deliberate the biological significance of DEGs, DEGs were analyzed using GSEA of Oncogenic signatures and CGP. (A) Up-regulated genes were enriched in the "VEGF_A_UP.V1_UP" term. (B) Up-regulated genes were enriched in the "P53_DN.V2_UP" term. (C) Down-regulated genes were enriched in the "VEGF_A_UP.V1_DN" term. (D) Up-regulated genes were enriched in the "KOBAYASHI_EGFR_SIGNALING_24HR_UP" term. (E) Up-regulated genes were enriched in the "YAN_ESCAPE_FROM_ANOIKIS" term. (F) Up-regulated genes were enriched in the "KOBAYASHI_EGFR_SIGNALING_24HR_DN" term.

## Bypass signaling in EGFR pathway may influence the cell cycle

Some up- or down-regulated genes were enriched in the terms "KOBAYASHI_EGFR_ SIGNALING_24HR _UP" and "KOBAYASHI_EGFR_SIGNALING_24HR_DN" respectively, both terms were related to EGFR inhibition (*Kobayashi et al., 2006*). In addition, our RT-PCR results showed that the expression of EGFR was down-regulated in CAL27[AR] cells (Fig. 5A). Several reports have demonstrated that the EGFR signaling pathway inhibition could be offset by activation of bypass signaling within the pathway, such as that involved in the ERBB, IL-6, PDGF and VEGF signaling pathway (*Akhavan et al., 2013*; *Cretella et al., 2014*; *Kim et al., 2012a*; *Viloriapetit et al., 2001*). Moreover, some studies have revealed that IL-6 and HGF could help cancer cells to acquire the ability of anoikis resistance (*Fofaria & Srivastava, 2014*; *Zeng et al., 2002*), suggesting bypass signaling in the EGFR pathway may play a significant role in anoikis resistance. Thus, we detected IL-6, PDGF and VEGF expression in CAL27[AR] cells. In contrast to the reduced expression of IL-6 in CAL27[AR] cells (Fig. 5A), the expression of VEGFA and PDGFA were up-regulated in CAL27[AR] cells (Fig. 5A). Furthermore, we confirmed that two gene

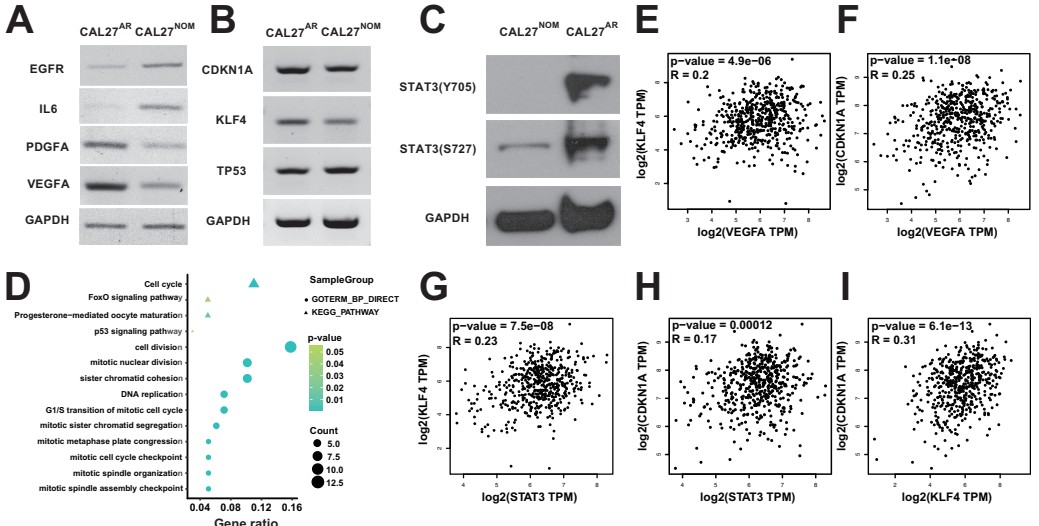

**Figure 5** **VEGFA-STAT3-KLF4-CDKN1A signal axis may be activated in CAL27<sup>AR</sup> cells.** (A) RT-PCR showing that EGFR and IL-6 were down-regulated in CAL27<sup>AR</sup> cells. PDGFA and VEGFA were up-regulated in CAL27<sup>AR</sup> cells. (B) RT-PCR showing that CDKN1A and KLF4 were up-regulated in CAL27<sup>AR</sup> cells. The expression of TP53 was down-regulated in CAL27<sup>AR</sup> cells. (C) Western blotting analysis showing that the expression ofphospho-STAT3 (Tyr 705) proteincould only be detected in CAL27<sup>AR</sup> cells. Both CAL27<sup>AR</sup> and CAL27 cells could express phospho-STAT3 (Ser 727) protein. (D) The function analysis of down-regulated genes related to the term "VEGF_A_UP.V1_DN" using DAVID. The results showing that overexpression of VEGFA might negatively influence cell cycle. These cell cycle processes and eventsincluded DNA replication and G1/S transition. (E)–(I) Gene correlation analysis in HNSCC patients. (E) The correlation between VEGFA and KLF4 was positive. (F) The correlation between VEGFA and CDKN1A was positive. (G) The correlation between STAT3 and KLF4 was positive. (H) The correlation between STAT3 and CDKN1A was positive. (I) The correlation between KLF4 and CDKN1A was positive.

sets related to VEGFA overexpression (*Schoenfeld et al., 2004*): "VEGF_A_UP.V1_UP" and "VEGF_A_UP.V1_DN", were enriched in up- or down-regulated genes (Figs. 4A and 4C), respectively, which indicated that VEGF signaling was activated. To explore the function of VEGFA in CAL27<sup>AR</sup> cells, the DEGs related to the "VEGF_A_UP.V1_UP" and "VEGF_A_UP.V1_DN" terms were further analyzed using DAVID. It was showed that the genes down-regulated by VEGFA overexpression negatively influenced the cell cycle, especially DNA replication and G1/S transition (Fig. 5D). This result was consistent with the GO enrichment analysis and GSEA of Reactome pathways that were mentioned. Thus, our results suggested that the activation of the VEGFA signaling pathway blocked the cell cycle progression in CAL27<sup>AR</sup> cells.

## VEGFA-STAT3-KLF4-CDKN1A signal axis presents in multiple epithelial tumors with high expression of VEGFA

A recent study reported by *Zhao et al. (2015)* demonstrated that VEGFA could promote phosphorylation of STAT3 and increase KLF4 expression. Thus, we then detected the expression of phospho-STAT3 (Tyr 705) and phospho-STAT3 (Ser 727) protein in CAL27<sup>AR</sup> cells using western blotting. Our results showed that both CAL27<sup>AR</sup> and CAL27
cells could express phospho-STAT3 (Ser 727) protein. However, the expression of phospho-STAT3 (Tyr 705) protein could only be detected in CAL27^AR cells (Fig. 5C). *Hall et al. (2009)* reported that transcription of Krüppel-like transcription factor 4 (KLF4) could be activated by phospho-STAT3 (Tyr 705). KLF4 could block cell cycle progression via up-regulating CDKN1A expression and repressing TP53 transcription, which produces an anti-apoptotic effect (*Rowland & Peeper, 2006*). More importantly, the cell cycle arrest mediated by KLF4 might be the outcome of inhibition of G1/S transition, as KLF4 could activate CDKN1A transcription (*Chen et al., 2001*; *Wei et al., 2006*). For this reason, we detected KLF4, TP53 and CDKN1A expression in CAL27^AR cells. Our results showed that the CDKN1A and KLF4 expression was up-regulated in CAL27^AR cells (Fig. 5B), which is in accordance with the results of RNA-Seq. Although some reports showed that CDKN1A could induce G1/S transition arrest and that its expression is regulated by TP53 (*Cazzalini et al., 2003*; *El-Deiry et al., 1994*; *Riley et al., 2008*), TP53 expression was down-regulated in CAL27^AR cells (Fig. 5B). In addition, our enrichment results indicated that the p53 signaling pathway was suppressed in CAL27^AR cells as some up-regulated genes were enriched in the term ''P53_DN.V2_UP'' (Fig. 4B)—a gene set related to TP53 silence (*Elkon et al., 2005*). Consequently, the overexpression of CDKN1A might be induced by KLF4 in CAL27^AR cells rather than TP53. Thus, it is possible that VEGFA-STAT3-KLF4-CDKN1A signal axis might be present in CAL27^AR cells. To verify this view, based on TCGA database, gene correlation analysis was used to detect correlations among VEGFA, STAT3, KLF4 and CDKN1A genes. The results of gene correlation analysis revealed that not only VEGFA expression was positively related to KLF4 and CDKN1A expression in HNSCC (Figs. 5E and 5F), but also that STAT3 was positively related to KLF4 and CDKN1A expression (Figs. 5G and 5H). In addition, the correlation between KLF4 and CDKN1A genes was also positive (Fig. 5I) in HNSCC. Overall, these results confirmed the fact that VEGFA-STAT3-KLF4-CDKN1A signal axis might be present in CAL27^AR cells.

To test whether the VEGFA-STAT3-KLF4-CDKN1A signal axis was accidental in HNSCC, we studied the cancers with high expression of VEGFA based on TCGA database. Data from TCGA database indicated that VEGFA is also expressed highly in kidney renal clear cell carcinomas (KIRC), ovarian serous cystadenocarcinomas (OV) and glioblastoma (GBM), as well as HNSCC (Figs. 6A and 6B). Further, we studied the correlation among VEGFA, STAT3, KLF4 and CDKN1A genes in KIRC, OV and GBM, with the results showing that the correlation among VEGFA, STAT3, KLF4 and CDKN1A genes in KIRC and OV was consistent with that in HNSCC (Figs. 7A–7J). This indicated that the VEGFA-STAT3-KLF4-CDKN1A signal axis not only exists in the HNSCC, but also in the two other kinds of epithelial cancers that showed high expression of VEGFA. However, the relationship between VEGFA and KLF4 genes have no statistical significance in glioblastoma (Fig. 7O). Therefore, whether the VEGFA-STAT3-KLF4-CDKN1A signal axis existed in glioblastomas remains unclear. Thus far, the study has argued that VEGFA-STAT3-KLF4-CDKN1A signal axis may play an important role in epithelial cancers with high expression of VEGFA, such as HNSCC, KIRC and OV.

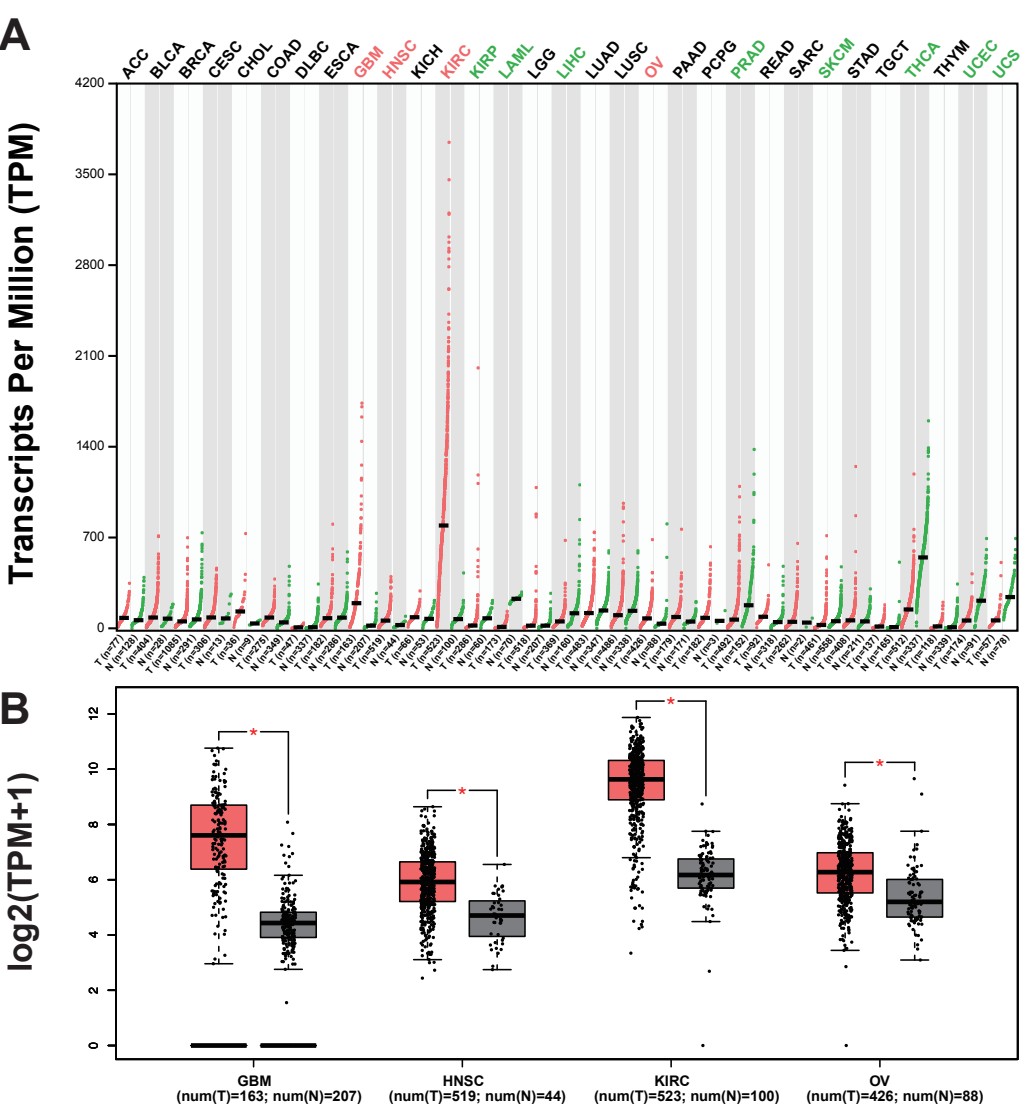

**Figure 6** **Cancer type that highly expressed VEGFA gene.** We studied the cancer type with high expression of VEGFA based on the TCGA database using GEPIA. (A) The expression of VEGFA in different tumor tissues. (B) The comparison of VEGFA expression among tissues of HNSCC, KIRC, OV, GSM and corresponding normal tissues. The results revealed that VEGFA also was expressed highly in kidney renal clear cell carcinoma (KIRC), ovarian serous cystadenocarcinoma (OV) and the glioblastoma (GBM), in addition to HNSCC.

# DISCUSSION

## Exosome released by CAL27[AR] cells might play a significant role in cancer metastasis

GO enrichment analysis revealed that the most significant up-regulated genes (Loget $\geq$ 2, probability $\geq$ 0.9) are related to cell differentiation, and the most significantly down-regulated genes (Loget $\leq$ −2, probability $\geq$ 0.9) are closely related to the cell cycle (Figs. 3A and 3B). This indicated that active cell differentiation as well as relatively stable cell

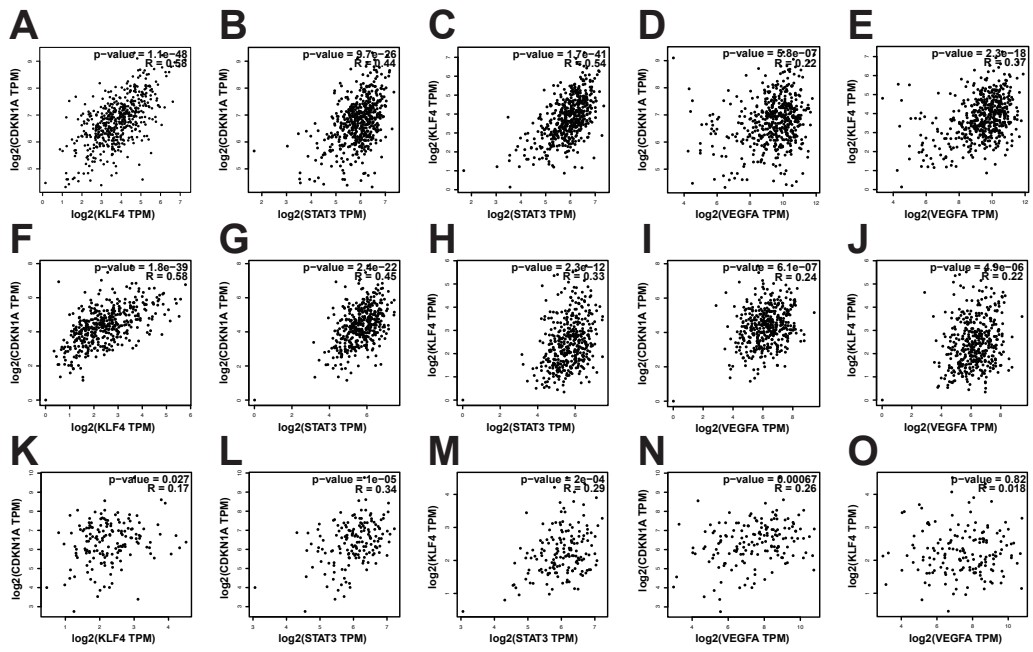

**Figure 7  Gene correlation analysis of KIRC, OV and GBM.** The gene correlations among VEGFA, STAT3, KLF4 and CDKN1A in KIRC ,OV and GSM were analyzed using GEPIA. (A–E) The result of gene correlation analysis of KIRC. (F–J) The result of gene correlation analysis of OV. (K–O) The result of gene correlation analysis of GBM. The results showed that there was a positive correlation between KLF4 and CDKN1A in KIRC, OV and GBM. STAT3 had a positive correlation with KLF4 and CDKN1A, respectively, in KIRC, OV and GBM. In addition, VEGFA also has a positive correlation with KLF4 and CDKN1A, respectively in KIRC and OV ($p$-value ≤ 0.05). However, there is no statistically significant correlation between VEGFA and KLF4 in GBM.

quiescence exist in CAL27[AR] cells. This phenomenon is similar to the properties of cancer stem cells (CSCs) (*Clarke et al., 2006*; *Glauche et al., 2009*). *Chen et al. (2012)* reported that cancer cells could be induced to generate a special cell group that have properties of CSCs through the suspension culture. CSCs have a strong ability to secrete exosomes (*Hannafon & Ding, 2015*), which could explain why there are a large number of up-regulated genes associated with extracellular exosomes in CAL27[AR] cells. These up-regulated genes may contribute to the formation of tumor metastatic microenvironments. For example, *Ji et al. (2013)* demonstrated that S100A8 and S100A9 was overexpressed in exosomes, released by SW620 cells, an isogenic human colorectal cancer cell line. S100A8/A9 are regarded as promotors for cancer metastasis in multiple types of cancers, such as colorectal cancer, prostate cancer, gastric cancer and breast cancer (*Hermani et al., 2006*; *Ji et al., 2013*; *Yin et al., 2013*; *Yong & Moon, 2007*). In addition, a report has shown that the WNT signaling pathway has an essential role in maintaining stem and cancer stem cells (*Holland et al., 2013*). Furthermore, *DiMeo et al. (2009)* stated that WNT signaling could drive tumor metastasis through mediating cancer cell self-renewal. Moreover, Rebecca et al. demonstrated that WNT signaling is activated in patient-derived metastatic cancer stem-like cells (*Lamb et al., 2013*). Importantly, there is evidence that Wnt4, derived from

the exosomes of colorectal cancer (CRC) cells, could promote the migration of cancer cells and angiogenesis by activating $\beta$-Catenin signaling (*Lamb et al., 2013*). Furthermore, *Cheah et al. (2015)* stated that CD14-high bladder cancer (BC) cells could establish and maintain an immune suppressed, inflammatory tumor microenvironment, which could be beneficial for bladder cancer metastasis. They demonstrated that CD14 is a key gene with pro-tumorigenic effects in bladder cancer. Overall, these cases demonstrated that the up-regulated genes related to exosome might have a critical role in cancer metastasis in CAL27[AR] cells (Table S2).

### Abnormal cellular metabolic pathways can activate the PPARs signaling pathway, thereby enhancing the ability of anoikis resistance in CAL27[AR] cells

Peroxisome proliferator-activated receptors (PPARs), a member of steroid receptor superfamily, are ligand activated transcription factors (*Issemann & Green, 1990*). At present, there are three different subtypes of PPARs, namely PPAR α, PPAR β/ δ and PPAR γ (*Berger & Moller, 2002*). These three subtypes are activated by endogenous ligands and participate in the regulation of glucose, amino acid and lipid metabolism (*Sertznig et al., 2007*). At present, high expression of PPAR γ was found in primary squamous cell carcinomas and squamous cell carcinoma cell lines. Moreover, PPAR γ could enhance the ability of anoikis resistance in squamous cell carcinoma cells (*Masuda et al., 2005*). In addition, Shen et al. found that oleic acid (OA) promotes the expression and autocrine regulation of angiopoietin like 4 (ANGPTL4) protein through the PPARs signaling pathway. High expression of ANGPTL4 could also increase the anoikis resistance ability and promote cancer metastasis (*Shen et al., 2017*). In this study, we also found that the PPARs signaling pathway is activated in CAL27[AR] cells (Fig. 3A). Moreover, the PPARs signaling pathway is accompanied with activation of abnormal cell metabolism signaling pathways in CAL27[AR] cells, such as arachidonic metabolism and linoleic metabolism signaling pathways (Fig. 3A). It has been reported that the oxidized metabolites of arachidonic acid and linoleic acid could be conducted with the PPAR signaling pathway activation (*Chawla et al., 2001*; *Sheldrick et al., 2007*; *Viswakarma et al., 2010*). In addition, studies have shown that arachidonic acid could promote prostate cancer cell growth (*Vainio et al., 2011*). It is also reported that linoleic acid could promote breast cancer cell migration (*Byon et al., 2009*). Importantly, we also observed increased ANGPTL4 expression in CAL27[AR] cells (data not shown). From this context, we inferred that abnormal cell metabolism was activated to adapt to the environment of the suspension culture, which was supported by several reports (*Christofk et al., 2008*; *Davison et al., 2013*; *Schafer et al., 2009*) These abnormal cell metabolisms could activate the PPARs signaling pathway. The activation of PPARs signaling pathway could enhance the ability of anoikis resistance through increasing ANGPTL4 expression in CAL27[AR] cells.

### G1/S transition inhibition could provide a protection from anoikis for cancer cells with high expression of VEGFA

*Reginato et al. (2003)* showed that the expression of EGFR was down-regulated after epithelial cells detachment from ECM, as EGFR expression was dependent on the matrix

signals. We also found that EGFR expression was lower in CAL27^{AR} cells than CAL27 cells (Fig. 5A). In addition, results from GSEA of CGP indicated that two gene sets related to EGFR inhibition were enriched in up- or down-regulated genes, respectively (Figs. 4A and 4C). Thus, we inferred that the EGFR signaling pathway was suppressed in CAL27^{AR} cells. According to our knowledge, the EGFR signaling pathway could be inhibited by tyrosine kinase inhibitors (TKIs). However, the inhibition of the EGFR signaling pathway could be reversed by other signaling pathways, such as VEGFA signaling pathway (*Viloriapetit et al., 2001*). Indeed, we found that VEGFA was up-regulated in CAL27^{AR} cells (Fig. 5A). In addition, according to the results from the GSEA of oncogenic signatures, two gene sets related to VEGFA overexpression both were enriched in up- or down-regulated genes respectively (Figs. 4D and 4F). Thus, we supposed that the VEGFA signaling pathway might be activated in CAL27^{AR} cells. So we inferred that the VEGFA signaling pathway, as a bypass signaling pathway in the EGFR signaling pathway, may play an important role in resisting the EGFR signaling pathway inhibition in CAL27^{AR} cells. Moreover, we found that many down-regulated genes followed by VEGFA overexpression could influence G1/S transition (Fig. 5D). Both GO analysis and GSEA analysis revealed that G1/S transition was suppressed because many down-regulated genes were significantly enriched in the "G1/S transition" term (Fig. 2A, Table S3), which suggested that G1/S transition inhibition is a significant event in CAL27^{AR} cells.

*Collins et al. (2005)* stated that the G1/S transition inhibition could contribute to anoikis resistance in epithelial cells. We supported that G1/S transition inhibition might also be vital for anoikis resistance in CAL27^{AR} cells. As is known to us, CDKN1A (the gene that encodes p21) could block G1/S transition through inhibiting cyclin/Cdk complexes (*Gartel & Tyner, 2002*). We found that CDKN1A expression is up-regulated in CAL27^{AR} cells (Fig. 5B). *Gartel & Tyner (2002)* stated that CDKN1A expression is divided into p53-dependent and p53-independent mechanisms. As previously described, we found that overexpression of CDKN1A is a result of KLF4 activation rather than TP53 regulation in CAL27^{AR} cells. In this case, there is no doubt that both KLF4 and CDKN1A could help CAL27^{AR} cells to acquire anoikis resistance through G1/S transition inhibition. Moreover, we found that KLF4 transcription might be induced by phospho-STAT3 (Tyr 705) protein because phospho-STAT3 (Tyr 705) protein has been detected in CAL27^{AR} cells. Several studies revealed that STAT3 could enhance resistance to anoikis in pancreatic cancer cells, esophageal squamous cells and melanoma cells (*Du et al., 2009*; *Fofaria & Srivastava, 2014*; *Fofaria & Srivastava, 2015*). In this case, the STAT3-KLF4-CDKN1A signaling axis might be activated for enhancing anoikis resistance in CAL27^{AR} cells. We also demonstrated that VEGFA might be an activator of STAT3 phosphorylation in CAL27^{AR} cells. This perspective was verified by gene correlation analysis that VEGFA expression was positively related to KLF4 and CDKN1A expression (Figs. 5E and 5F). Importantly, we demonstrated that VEGFA overexpression might lead to G1/S transition inhibition (Fig. 5D). This result was in accordance with the effect of CDKN1A and KLF4 genes. Overall, cell cycle arrest at the G1 phase might prevent CAL27^{AR} cells from anoikis through activating VEGFA-STAT3-KLF4-CDKN1A signaling axis. We have proven that VEGFA expression is up-regulated in KIRC, OV and HNSCC tissues; furthermore, the VEGFA-STAT3-KLF4-CDKN1A signal

axis also presents in these three kinds of epithelial cancers tissues. Our study findings demonstrate that the VEGFA-STAT3-KLF4-CDKN1A signaling axis might have a role in anoikis resistance in KIRC, OV and HNSCC.

### Entry into quiescence driving by CDKN1A protected CAL27[AR] cells from anoikis

The suppression of cell cycle signaling pathway in CAL27[AR] cells, which was confirmed by GSEA, is an interesting phenomenon in the induction of anoikis resistance. Besides its role in cell metabolism, PPARs signaling pathway could inhibit cell cycle through its function in transcriptional regulation (*Altiok, Xu & Spiegelman, 1997*; *Miller, Wuertz & Ondrey, 2018*; *Toyota et al., 2002*). *Chung et al. (2002)* put that activation of PPAR-γ were able to induce cell cycle arrest in anaplastic thyroid cancer cells via a p53-independent, but CDKN1A dependent cytostatic pathway. Interestingly, we found that the VEGFA-STAT3-KLF4-CDKN1A signaling axis was activated during CAL27[AR] cells detachment from ECM, which could result in the G1/S transition inhibition of CAL27[AR] cells. Moreover, CDKN1A has been demonstrated to be involved in the stemness acquirement of cancer stem cells (*Kim & Singh, 2019*), which could explain the plenty of exosomes excreted by CAL27[AR] cells. Therefore, we propose that CDKN1A is a key regulator of anoikis resistance in CAL27[AR] cells.

*Perucca et al. (2009)* put that high CDKN1A levels are indispensable for cells to enter and maintain the quiescence state. Quiescence is an actively maintained state rather than a passive state lacking proliferative activities, which could provide protection against cellular stress (*Cheung & Rando, 2013*; *Coller, Sang & Roberts, 2006*; *Sang, Coller & Roberts, 2008*; *Yao, 2014*). *Daubriac et al. (2009)* showed that quiescent pluricellular aggregates could exhibit anoikis resistance property in malignant pleural mesothelioma, which supports our proposal that cell quiescence induced by CDKN1A might protect CAL27[AR] cells from anoikis.

## CONCLUSIONS

The acquirement of quiescence regulated by CDKN1A is critical for CAL27[AR] cells survival from anoikis. In addition, CAL27[AR] cells may release several exosomes that have a pro-oncogenic role in HNSCC metastasis.

### Funding

This work was supported by the Hubei Provincial Natural Science Foundation of China (No. 2016CFA067). The funders had no role in study design, data collection and analysis, decision to publish, or preparation of the manuscript.

### Grant Disclosures

The following grant information was disclosed by the authors:
Hubei Provincial Natural Science Foundation of China: 2016CFA067.

## Competing Interests

The authors declare there are no competing interests.

## Author Contributions

- Chen Guo and Ling-feng Xu performed the experiments, analyzed the data, prepared figures and/or tables, authored or reviewed drafts of the paper.
- Hui-min Li and Wei Wang performed the experiments.
- Ji-hua Guo contributed reagents/materials/analysis tools, authored or reviewed drafts of the paper.
- Meng-qi Jia prepared figures and/or tables.
- Rong Jia conceived and designed the experiments, performed the experiments, analyzed the data, contributed reagents/materials/analysis tools, authored or reviewed drafts of the paper, approved the final draft.
- Jun Jia conceived and designed the experiments, analyzed the data, contributed reagents/materials/analysis tools, authored or reviewed drafts of the paper, approved the final draft.

## Data Availability

The raw data are available at the National Center for Biotechnology Information Sequence Read Archive (SRA): accession number SRP158985.

All the full-length uncropped blots pictures can be downloaded from figshare: Guo, Chen; Jia, Jun; Xu, Ling-Feng; Li, Hui-Min; Wang, Wei; Guo, Ji-Hua; et al. (2019): Transcriptomic study of the mechanism of anoikis resistance in head and neck squamous carcinoma. figshare. Fileset. https://doi.org/10.6084/m9.figshare.7390229.v1.

## Supplemental Information

Supplemental information for this article can be found online at http://dx.doi.org/10.7717/peerj.6978#supplemental-information.

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
