# Peer review of "Transcriptomic study of the mechanism of anoikis resistance in head and neck squamous carcinoma"

_PeerJ, doi:10.7717/peerj.6978_

## Round 0.1 · original submission · Major Revisions

The authors need to make the writing more clear and logical.

Reviewer 1 ·

Basic reporting

No comment

Experimental design

No comment

Validity of the findings

No comment

Additional comments

The author presented a combination of computational and experimental work for the mechanism of anoikis resistance in head and neck squamous carcinoma. This is a good design and the results were validated by experiments. The topic is novel, however I do have a few concerns:

1. Your methods needs more detail. I suggest that you improve the description at lines 97- 100 to explain "How the author obtain the DEGs?"
2. I would also suggest the author make you result subtitle more specific. Like line 187 "Gene Set Enrichment Analysis (GSEA) of DEGs", could you please say what is the main purpose or finding from here.
3. Seems the authors have done many analysis, but not link to each other. It will be better to make a workflow for describe the relationship between different analysis restuls.
4. For Figure 1B, why all the RT-PCR is for normal, how about the AR cases?
5. For Figure 1A, what is the meaning for the color? FPKM?
6. In supplementary table, seems come Chinese charaters to be fixed.

·

Basic reporting

In this manuscript, the authors did RNA-seq of three CAL27 samples and three anoikis-resistant CAL27 samples induced by suspension culture. They did GO and KEGG pathway enrichment analysis on differentially expressed genes (DEGs), and reported that the down-regulated genes in anoikis-resistant CAL27 samples were enriched in cell cycle, while up-regulated genes enriched in extracellular exosome, and up- or down-regulated genes were accordantly related to VEGF and EGFR pathway. They did RT-PCR experimental validation of some differentially expressed genes, and then focused on validating VEGFA-STAT3-KLF4-CDKN1A signal axis.
In general, I thought this study is good, but the English and some logic may need to be improved. The authors seem to fail to mention some related literature references in introduction or compare with them in discussion, for example PubMed ID 22154512, 16007176. The article structure is professional. Figures and tables are mostly good and self-contained. Raw data can be found in SRA (SRP158985).

Experimental design

This study is within the scope of PeerJ. The research question is relevant and meaningful, but not very sharply defined in the Introduction section. Analysis and Results look good, although some methods may need more details.

Validity of the findings

The RNA-seq experiment was done on three anoikis-resistant CAL27 samples vs three control CAL27 samples. Differential expressed genes (DEGs) are defined with the standard that is not very clear and is changed several times, and I am not sure about the statistical significance of DEGs. Multiple test correction (FDR or Bonferroni or q-value) was not done for the first part of GO enrichment analysis by DAVID. The conclusion part may be a little more than that the results can support.

Additional comments

Some specific comments are listed below.
Line 34, 'though' should be a typo of 'through', in Results part in Abstract
In first paragraph of Introduction (Line 43-54), there should be a sentence to link the relationship between lip, oral cavity, pharyngeal cancer and HNSCC.
The sentence introducing anoikis (Line 51-52) should be moved ahead, before previous sentence mentioning 'cancer cells undergo anoikis'.
Line 52, 'a few cancer cells are prone to development of the ability to resist anoikis' might be better as 'a few cancer cells are likely to develop the ability to resist anoikis'
Line 57, 'allows transformed cancer cells survival' might be better as 'allows transformed cancer cells to survive'.
Line 64, 'mechanism' is a strong word, which cannot be made by bioinformatics analysis of association.
Line 66-67, 'RNA-Seq technique was used to obtain deep transcriptional analysis between…', RNA-seq is a sequencing technique, which is different from RNA-seq data analysis. And I think 'deep transcriptional analysis' looks strange, which might be better as 'deep sequencing data for whole transcriptome analysis'.
Line 70, the word 'meticulously' has fuzzy meaning; I am not sure what 'the key signaling pathways' refers to; 'discovered' might be strong.
Line 95, I am not sure what 'as described previously' refers to
Line 97-98, what are 'Loget' and 'probability'? Are the DEGs statistically significant?
Line 102 & 165, why the standard of DEGs is changed again?
Line 104-105 & 169-186, 'p-value<0.05' & 'p-value=…', why not use FDR or Bonferroni corrected p-value? Do the authors apply multiple test correction to the results of DAVID enrichment analysis. BTW, I am not sure if the scientific notation of 1.01E-21 should be written as 1.01*10^-21 (-21 is superscript) in the text.
Line 106, 'three ontologies of GO were collected picked up separately', two verbs next to each other.
Line 117, 'Different gene sets in the MSigDB were used to detect activated or repressive pathways related to our DEGs.' Why use different gene sets?
Line 123, 'GEPIA' needs citation, which seems to be Line 133-134.
Line 142, Table S1's column names are in Chinese and should be translated into English.
Line 159 & Figure 1 legend, 'had good heterogeneity and repeatability' might be better as 'had low heterogeneity and high repeatability'.
Line 160, 'All the detected 21 genes matches the results of RNA-Seq,' How are these 21 experimentally validated genes were selected? randomly or manually selected? Some of the 21 validated genes cannot be found in Fig S1, although can be found in Supplemental_file_S1.
Line 173, 'p-value = 2,63E-12' might be a typo of '2.63E-12'
Line 182, Supplement Table 2, why some gene length has fractions? why not also list genes in 'nucleoplasm', 'cell division', etc.?
Line 191, 'These signaling pathways included' refers to 'activated signaling pathways'?
Line 196, since cell cycle processes were suggested to be suppressed in CAL27AR cells, do CAL27AR cells grows more slowly?
Line 201-202, one unnecessary line break
Line 220, 'DEGs enriched in … terms', do you mean 'DEGs related to … terms'? because enrichment should be on the level of function/pathway terms, but not genes.
Line 222, 'negatively influenced the cell cycle', do you mean the genes down-regulated by VEGFA overexpression, or VEGFA overexpression may suppress cell cycle?
Lines 254, 'First,' I did not see any 'second' in latter part. And this paragraph seems to be an extension of the signal axis to other cancer types, but not 'To further verify our view,' (Line 253).
Line 357-358, in Conclusions section, 'Abnormal cellular metabolism changes … may both result in anoikis resistance'. There seems to be no experiments to show there are actually abnormal cellular metabolism changes. And the bioinformatics analysis and validation on DEGs may only show association but not causal relationship with 'result in'.
In Figure 1A, the number -2~2 with red~black~green bar in the legend, what is the meaning of the number? z-score? BTW, red-green heatmap might be unfriendly to red-green color blind readers.
In Figure 1B, how are these validated genes were selected? I did not fin KRT80, KRT8, KRT18, KT5, LRRN4 in Fig S1.
In Figure 2B, 'serine-type endopeptidase inhibitor activity' seems to have FDR > 0.05 when I rerun DAVID.
In Figure 2C, the legend for 'Count' is better to have some more levels.
I did not find figure legend for supplemental figures.
As previously mentioned, some related literature references, for example PubMed ID 22154512, 16007176, might be mentioned in introduction or compare with their results in discussion. The purpose and research question might be better sharply defined in introduction section.

---

## Round 0.2 · Minor Revisions

I can see the author make a lot of changes, however please answer some reviewers' concerns, including those related to the the figures and the discussion on different results from different sections.

Reviewer 1 ·

Basic reporting

No comments

Experimental design

No comments

Validity of the findings

No comments

Additional comments

The author made some revision, however, some important comments not addressing well. Like my previous comment#3, how could author link different results. The author give a workflow, which is good. My point is how different results can support each other for a whole story, like any interesting genes shared in different analysis results. I would hope the author could add some discussion on this.

For figure 1A, it is still hard to recognise the expression difference, I would suggest change you colour index and make it more clear.

One more question is about Figure 1B, like the ALOX5AP, how could the result is a lot of spot instead of bands. It need more explanation, it is contamination or not.

·

Basic reporting

no comment

Experimental design

no comment

Validity of the findings

no comment

Additional comments

The authors have made great efforts to improve the manuscript, and have resolved most of my concerns.

A minor comment is that some answers in the rebuttal letter seem not to have corresponding revision in the manuscript. To make the manuscript clearer, it might be better to make the revision.
1) All supplemental figures might be better to be briefly introduced and cited in the main text, including the new supplemental figure "workflow". And figure legends for supplemental figures still cannot be found (I mean the introductory text figure legends, like the page before each main-text figure).
2) "Loget" and "probability" (of difference) is not well defined or explained or cited to other reference in Methods section.
3) The selection of 21 RT-PCR validated genes (15 keratin-related genes + 6 randomly selected other genes) is not introduced or mentioned at all in either Methods or Results section.
4) The explanation of the color legend in Figure 1A - "z-score" (transformed FPKM) - is still not shown anywhere, which could be either as the title of the color legend or in the text figure legend.

---

## Round 0.3 · accepted · Accept

This revision make the idea much clearer.

#